# *ApoA1* Deficiency Reshapes the Phenotypic and Molecular Characteristics of Bone Marrow Adipocytes in Mice

**DOI:** 10.3390/ijms23094834

**Published:** 2022-04-27

**Authors:** Afroditi Kastrenopoulou, Kyriakos E. Kypreos, Nicholaos I. Papachristou, Stavros Georgopoulos, Ioulia Mastora, Ioanna Papadimitriou-Olivgeri, Argyro Spentzopoulou, Dragana Nikitovic, Vassilios Kontogeorgakos, Harry C. Blair, Dionysios J. Papachristou

**Affiliations:** 1Laboratory of Bone and Soft Tissue Studies, Department of Anatomy-Histology-Embryology, Unit of Bone and Soft Tissue Studies, School of Medical, University of Patras, 26504 Patras, Greece; afroditika@gmail.com (A.K.); n_papaxristou@yahoo.gr (N.I.P.); stavrosbuzuki@hotmail.gr (S.G.); ioulia.mastora@gmail.com (I.M.); ioannaolivgeri@gmail.com (I.P.-O.); argspent@gmail.com (A.S.); 2Department of Pharmacology, School of Medical, University of Patras, 26504 Patras, Greece; kkypreos@upatras.gr; 3Laboratory of Department of Anatomy-Histology-Embryology, School of Medical, University of Crete, 71110 Heraklion, Greece; nikitovic@uoc.gr; 4Department of Orthopeadic Surgery, School of Medicine, University of Athens, 11527 Athens, Greece; vaskonto@gmail.com; 5Pittsburgh VA Medical Center, Pittsburgh, PA 15261, USA; 6Department of Pathology, University of Pittsburgh, Pittsburgh, PA 15261, USA

**Keywords:** apolipoprotein A-1, high-density lipoprotein, brown adipose tissue, white adipose tissue, beige (hybrid) adipose tissue

## Abstract

In the present study, we studied the effect of apolipoprotein A-1 (APOA1) on the spatial and molecular characteristics of bone marrow adipocytes, using well-characterized *ApoA1* knockout mice. APOA1 is a central regulator of high-density lipoprotein cholesterol (HDL-C) metabolism, and thus HDL; our recent work showed that deficiency of APOA1 increases bone marrow adiposity in mice. We found that *ApoA1* deficient mice have greatly elevated adipocytes within their bone marrow compared to wild type counterparts. Morphologically, the increased adipocytes were similar to white adipocytes, and displayed proximal tibial-end localization. Marrow adipocytes from wild type mice were significantly fewer and did not display a bone-end distribution pattern. The mRNA levels of the brown/beige adipocyte-specific markers *Ucp1*, *Dio2*, *Pat2*, and *Pgc1a*; and the expression of leptin were greatly reduced in the *ApoA1* knock-out in comparison to the wild-type mice. In the knock-out mice, adiponectin was remarkably elevated. In keeping with the close ties of hematopoietic stem cells and marrow adipocytes, using flow cytometry we found that the elevated adiposity in the *ApoA1* knockout mice is associated with a significant reduction in the compartments of hematopoietic stem cells and common myeloid, but not of the common lymphoid, progenitors. Moreover, the ‘beiging’-related marker osteopontin and the angiogenic factor VEGF were also reduced in the *ApoA1* knock-out mice, further supporting the notion that APOA1—and most probably HDL-C—regulate bone marrow microenvironment, favoring beige/brown adipocyte characteristics.

## 1. Introduction

Mammalian adipose tissue is classified in two general categories, white and brown adipose tissue, which have striking differences as regards their origin, histological features, molecular characteristics, and functions [1]. White adipose tissue (WAT) is composed of large, unilocular, lipid containing cells. It is primarily located subcutaneously and around visceral organs. It serves as a pool that stores excessive energy, which is released in the form of fatty acids when required. Conversely, brown adipose tissue (BAT) is comprised of multilocular, mitochondria-enriched cells. It is encountered at scapulae, along the great vessels and in the retroperitoneum of rodents and humans. ΒAΤ has cardinal role in the heat-generating process, thermogenesis. The theromogenic capacity of brown fat cells is primarily attributed to the abundance of tightly packed mitochondria that contain the uncoupling protein 1 (UCP1), which, upon activation, short-circuits the respiratory chain, uncoupling respiration from ATP synthesis, thereby releasing chemical energy as heat [2]. Beyond UCP1, other factors—including type 2 iodothyronine deiodinase (DIO2), the UCP-1 transcription co-regulators PRDM16 and PGC1α, and the lipolysis modulator CIDEA—are part of the BAT-specific molecular cassette [3,4].

Under particular conditions, such as exposure to cold and adrenergic stimulation, WAT can acquire BAT-like characteristics, including UCP1 expression [1,5]. This hybrid type of fat tissue is referred to as ‘beige’ or ‘bright’ adipose tissue (BeAT). Similarly to WAT, BeAT originates from Myf5-precursors [1]; nevertheless, beige adipocytes are multilocular cells, rich in mitochondria that express UCP1 and other thermogenesis regulators. Therefore, BeAT adipocytes, morphologically and functionally, are closer to brown than to white adipocytes. Several lines of evidence suggest that BeAT exerts favorable effects on body weight, insulin sensitivity, and lipid metabolism. Thus, modulation of a ‘white’ to ‘beige’ switch has been proposed as a potential therapeutic approach against metabolic pathologies, obesity, the metabolic syndrome, diabetes, and others [5].

Bone marrow adipose tissue has relatively recently been acknowledged as a separate category of adipose tissue, together with WAT and BAT. In fact, the presence of fat cells within the bone marrow has attracted the interest of the scientific community for several decades. However, at that time, marrow adipocytes were mainly thought of as ectopic fat deposits that occupy marrow cavity, just replacing hematopoietic progenitors and other blood cells, hence indirectly affecting hematopoiesis [6]. Later, the close relationship—both anatomical and functional—between marrow fat and its microenvironment was noted and explored. 

It is now accepted that bone marrow adipose tissue is an active organ component with defined and significant metabolic functions. From a micro-morphological standpoint, bone marrow adipose tissue resembles WAT, since bone marrow adipocytes are unilocular and contain single lipid droplets that push nuclei towards the membrane periphery. Scheller et al. described two discrete types of bone marrow adipocytes in rodents [7]. The first are ‘constitutive’ adipocytes, present from birth and reside in vertebrae and distal extremities. The second are ‘regulated’ adipocytes that are more proximal; in close vicinity to red marrow; and respond to several environmental, hormonal, and nutritional cues [7].

Bone marrow adipose tissue shares molecular characteristics with both BAT and WAT. Indeed, bone marrow adipose tissue expresses the master regulators of lipoblastic differentiation PPARγ and CEBPa and the ‘white-like’ adipokines leptin and adiponectin, along with the ‘brown/beige-like’ markers DIO2, PRDM16, and PGC1α [8]. The expression of UCP1 in bone marrow adipose tissue and the role of bone marrow adipose tissue in thermogenesis are still under investigation. However, an increasing volume of data suggests a tight bond between bone marrow adipose tissue and lipid metabolism. Indeed, we recently showed that shortage of high-density lipoprotein (HDL) due to *ApoA1* deficiency culminates in increased bone marrow adiposity, as well as enhanced *Cebpa* and *PPARγ* expression, in mice [9]. However, the histomorphological and molecular characteristics of these adipocytes have not been investigated. Triggered by this, in the present study we examined the effects of *ApoA1* deficiency—and thus, impaired HDL biosynthesis—on the micro-morphology and molecular features of bone marrow adipocytes from mice tibiae. 

## 2. Results

### 2.1. Microscopic Characteristics of Bone Marrow Adipocytes of ApoA1 Knockout Mice

We previously showed that *ApoA1* KO mice exhibit significantly elevated bone marrow adipocytes in comparison to their WT counterparts [9]. Consistent with this, we confirmed that *ApoA1* deficiency results in augmented bone marrow adiposity (Figure 1A,B). Microscopically, the bone marrow adipose tissue comprised unilocular cells, containing a single lipid droplet that displaced nucleus towards cell periphery, closely resembling white adipocytes. A constant finding was that these cells were located at the proximal part of the examined tibiae (Figure 1A,B). On the other hand, most of the sections obtained from WT mice bones (tibiae and vertebra) had very few adipocytes that displayed WAT-type morphology, without obvious area-specific preferences (Figure 1C,D). The mean size of the bone marrow adipocytes was 30.815 μm (SD = 2.945) for the KO and 29.850 μm (SD = 3.297) for the *ApoA1* KO and the WT mice, respectively. These differences are not significant (*p* = 0.3351). Chronic inflammation or other remarkable microscopic features were not evident in sections from bones of all the tested animals. 

### 2.2. ApoA1 Deficiency Is Associated with Reduced Expression of the BAT-Specific Genes and Differential Expression of the Two Major Adipokines Leptin and Adiponectin

To determine whether *ApoA1* deficiency has any impact on molecular characteristics of bone marrow adipocytes, we examined the mRNA of specific genes that related to brown fat. Notably, *ApoA1* KO mice displayed strongly reduced expression of the BAT-like genes *Ucp1* (*p* = 0.05), *Pgc1a* (*p* = 0.03), *Pat2* (*p* = 0.005), and *Dio2* (*p* = 0.01) (Figure 2). The expression of the white fat associated adipokine leptin was also greatly decreased (*p* = 0.0001), in sharp contrast to adiponectin that displayed significant elevation (*p* = 0.012) (Figure 2). These findings support an active involvement of APOA1 in ‘shaping’ marrow adiposity, affecting the white-to-brown switch.

### 2.3. ApoA1 Paucity Affects the Hematopoietic Stem Cells and the Common Myeloid, but Not the Common Lymphoid Progenitors 

Flow cytometric analysis on whole bone marrow cells, revealed that the hematopoietic stem cells compartment (Lin-c-kit + Sca1+) were reduced in the *ApoA1*^-/-^, as compared to their WT counterparts (*p* = 0.002). Interestingly, the common myeloid progenitors compartment (Lin-c-kit low Sca1 low) was also greatly reduced (*p* = 0.001) in the *ApoA1* deficient mice, whereas the other hematopoietic stem cells (HSCs) branch—the common lymphoid progenitors compartment (Lin-c-kit + Sca1−)—was unaffected (Figure 3A,B). In line with the flow cytometry data, RT-PCR analysis uncovered that the expression levels of *Alcam1* (activated leukocyte cell adhesion molecule 1), a molecule that is expressed on primitive HSCs and activated lymphocytes and monocytes, were also very consistently decreased in the *ApoA1* KO mice (*p* < 0.0001) (Figure 3C).

### 2.4. ApoA1 Deficiency Reduces the Expression of Opn and Vegf, Genes Implicated in the WAT–BAT Switch 

Recent studies showed that the multifunctional protein osteopontin (*Opn*) promotes BAT synthesis from white pre-adipocytes via activation of the PI3K-AKT signal transduction pathway [10]. In the same context, several lines of evidence suggest that vascular endothelial growth factor (VEGF) participates in WAT vasculature regulation, promoting angiogenesis and WAT ‘beiging’ [11,12]. Triggered by this, in the present work, we examined the mRNA levels of *Opn* and *Vegf* in the WT and the *ApoA1* KO mice. We found that whole bone marrow cells obtained from *ApoA1* deficient mice exhibit robust *Vegf* mRNA downregulation (*p* = 0.02) and notable Opn reduction (*p* = 0.007), which, however, did not quite reach statistical significance (*p* = 0.076) (Figure 4). These findings suggest a possible role of these molecules in the ‘white-to-brown/beige’ transition. 

## 3. Discussion

This study further explored the reduced bone mass in C57BL/6 mice [9] by the footprint of APOA1 on the phenotype of bone marrow adipose tissue. Bone marrow adipose tissue is a separate category of adipose tissue that occupies approximately 10% of fat mass in adults [6]. Notably, bone marrow adipose tissue has morphological and molecular features of both WAT and BAT. Histologically, bone marrow adipocytes resemble white adipocytes: Spherical cells containing one triglyceride vacuole that pushes the nucleus towards the cell periphery; on average, their size is smaller than that of visceral and peripheral subcutaneous fat [13]. Marrow adipocytes of the *ApoA1* KO mice are similar to white adipocytes. Bone marrow of control mice has very few fat cells (Figure 1B). 

Bone marrow adipose tissue is a metabolically energetic organ that interacts with other marrow cells responding to paracrine, endocrine, and nutritional cues. Indeed, bone marrow adipose tissue is affected by aging, obesity, estrogen depletion, type 2 diabetes, anorexia nervosa, and lipodystrophies [7] that lead to unbalanced or attenuated bone remodeling. However, bone marrow adipose tissue functions are location-specific and therefore marrow adipocytes might have distinct roles in different bones or even different parts of the same bone. 

Two sub-categories of bone marrow adipose tissue are described in mice: the proximal ‘regulated’ (r-bone marrow adipose tissue) and the distal ‘constitutive’ (c-bone marrow adipose tissue). Constitutive bone marrow adipose tissue is sluggish and accumulates primarily in the distal skeleton. Regulated r-bone marrow adipose tissue has a predilection for vertebra and proximal limbs that are characterized by active hematopoiesis, bone remodeling, and enhanced expression of brown adipose tissue-related markers UCP1, DIO2, and PGC1A [13,14]. Our histological analysis of marrow adipocytes revealed that *ApoA1* deficiency shifts MSC towards a phenotype resembling r-bone marrow adipose tissue. Our hypothesis that bone marrow adipocytes from *ApoA1* KO mice mostly fall in the metabolically active r-bone marrow adipose tissue category is supported by our published findings showing that elevated bone marrow adiposity in these animals affects surrounding osteoblasts and bone remodeling [9].

It has been previously reported that bone marrow adipocytes are negative regulators of HSCs [15]. Driven by this, in the present study we examined how *ApoA1* KO mice with increased marrow fat might alter blood cells. Flow cytometry showed a significant reduction in the compartment of the HSC and the myeloid progenitors; the lymphoid progenitors compartment was unaffected. Consistent with this, *ApoA1* KO mice had reduced mRNA of Alcam1, expressed on primitive HSCs and essential for myeloid colony formation [16,17]. The aforementioned data offer further support to our hypothesis that APOA1—and most probably HDL deficiency—‘reshape’ the bone marrow microenvironment molecular characteristics. However, since these findings are descriptive further, in vitro studies are required to investigate the functionality of HSC in a background of APOA1 shortage.

We previously showed that *ApoA1* deficiency augments the lipoblastic master regulators CEBPa and PPARγ [9]. Since bone marrow adipose tissue shares molecular features with both WAT and BAT, we asked if the absence of *ApoA1* is followed by alterations in the expression of WAT-specific genes. The brown/beige fat-related genes Ucp1, Pgc1a, Pat2, and Dio2 were expressed in wild type C57BL/6 mice, in concert with previous studies [7]. However, the expression of these genes was significantly reduced in the bone marrow of *ApoA1* KO, as compared to WT mice. Obviously, since we used whole bone marrow cells, these findings describe the impact of *ApoA1* deficiency on the entire bone marrow transcriptome. However, they reflect the effect of *ApoA1* shortage on bone marrow adipocytes, since we have used primers for genes highly specific for white or brown adipocytes. In addition, previous studies have shown that the results obtained from whole bone marrow cells are very similar to those of in vitro studies, regarding bone marrow fat [9]. Our findings indicate that APOA1 may have a central role in the maintenance of ‘browning’ of bone marrow adipose tissue. Clearly, additional studies—both in vivo and in vitro—will define further this hypothesis. 

Leptin is a conserved adipocyte-derived secreted protein in mammals, amphibians, reptiles, and fish. It has a cardinal role in physiological processes including glucose homeostasis, energy intake, and body weight control. In vitro and in vivo data showed that leptin is expressed in BAT at lower levels than WAT [18]. Commins et al. showed that leptin reduces WAT through a UCP1-dependent peripheral mechanism in mice [19]. Recently, Wang et al. revealed that in C57BL/6 mice leptin promotes WAT browning via inhibition of the Hedgehog [20]. In the present work, we uncovered that mRNA levels of leptin are significantly reduced in the *ApoA1* KO mice, compared to wild type, contributing to the increased bone marrow adiposity observed in these animals. Suppressed leptin expression implies that APOA1, and most probably HDL, have a role in the acquisition and/or maintenance of brown/beige phenotype in bone marrow.

Adiponectin is another major adipokine secreted by adipose tissues with a key role in energy homeostasis. In contrast to leptin, adiponectin is decreased in obesity [21]. Notably, the amount of adiponectin produced by MSC-derived and primary bone marrow adipose tissue adipocytes is lower than that secreted by peripheral WAT, in humans and mice [22]. This is not the case in extreme conditions including anorexia nervosa or cancer, where bone marrow adipose tissue adiponectin expression surpasses WAT [23]. Studies in mice showed that adiponectin reduces BAT and UCP1 levels, affecting thermogenesis [24]. In symphony with these data, we found that in *ApoA1* KO mice with reduced brown/beige markers, expression of adiponectin (Adipoq) was increased remarkably. Adiponectin suppresses bone by modulation of RANKL/OPG [20], so our findings indicate that APOA1/HDL deficiency links marrow ‘browning’ to bone remodeling. 

In a recent in vitro study, Zhong et al. demonstrated that osteopontin promotes BAT adipogenesis from white pre-adipocytes via activation of the PI3K-AKT and or CD-44 dependent signaling axes [20,25]. Driven by this, in the present study we explored whether the expression levels of osteopontin display any differences between *ApoA1* KO and WT mouse groups. We found that the mRNA of osteopontin (Opn), was reduced in the bone marrow of *ApoA1* KO mice, compared to WT, although this reduction slightly missed significance (*p* = 0.076). This finding is not surprising, since *ApoA1* deficiency is associated with reduced osteoblastic differentiation and function [9] and proposes a potential role of osteopontin in the regulation of bone marrow microenvironment in a background of *ApoA1* deficiency that deserves further investigation. 

The vascular endothelial growth factor-A (VEGFA) participates in several physiological processes including vasculogenesis, angiogenesis, and tissue regeneration in adult mammals [26]. It is also established that VEGF is implicated in WAT vasculature regulation, promoting angiogenesis and consequently marrow fat beiging [27,28]. Notably, VEGFA upregulation is upstream to UCP1 and PGC1α activation and hence an immediate early event in the beiging process [29]. We show here that VEGFA transcription is greatly reduced in the *ApoA1* KO mice (Figure 4), paralleling suppressed BAT-related markers in the bone marrow of these mice. This fits well with the model that the UCP1 transcriptional co-activator PGC1α promotes VEGFA activation [2]. Together, these findings are novel evidence that VEGFA and related angiogenic pathways may serve as mechanistic link connecting APOA1 to ‘white-to-brown’ switch. In future work, additional studies at the protein level are planned. Cells from bone marrow flushing were not sufficient for Western blot, and antibody labeling was not possible in fixed mouse long bones. 

In summary, in the present study we explored the spatial, morphological, and molecular characteristics of bone marrow fat in WT and *ApoA1*^-/-^ mice, and provided novel evidence that *ApoA1*—and most probably HDL—may have a role in the processes of ‘white-to-brown’ switch. These findings add to the rapidly evolving field of bone marrow adiposity, and propose that APOA1 manipulation might be a promising target to combat fat-related metabolic disorders. 

## 4. Materials and Methods

### 4.1. Experimental Animals

Twelve-week-old male *ApoA1* deficient mice (*ApoA1*^-/-^) (n = 5), backcrossed on C57BL/6 10 generations, as well as wild type (WT) C57BL/6 mice (n = 5), were obtained from Jackson Labs, Bar Harbor, Maine, USA. C57BL/6 littermates were used as controls. In symphony with our previously published studies, herein we use only male mice in order to ensure similar metabolic background, given that bone is sensitive to even minimal endocrine changes [9,30,31]. Experimental animals were fed standard chow, 29% protein, 60% carbohydrates, 11% fat (Mucedola SRL, Milan Italy), ad libitum in a 12 h dark/light cycle (7:00 a.m.–7:00 p.m. light). Genotyping was performed by tail DNA PCR. Their average body weight, plasma cholesterol, triglycerides and glucose were similar, as we have previously described [30]. After 12 weeks, mice of all groups were euthanized and tibiae and lumbar vertebrae were isolated for further analyses in line with published standards [32]. All animal experiments strictly followed the EU guidelines for the Protection and Welfare of Animals. Sample size estimation was conducted via power analysis, using Stat UBC (http://www.stat.ubc.ca/~rollin/stats/ssize/n2.html, accessed on 19 January 2020). All experiments were performed at least three times; there were no excluded data. The study was evaluated and approved by the committee of the Laboratory Animal Centre of the University of Patras Medical School and the Veterinary Authority of the Prefecture of Western Greece. 

### 4.2. Histology

Following euthanasia, tibiae were removed and fixed in 4% formalin (Merck, Kenilworth, NJ, USA) overnight. Fixed tissue samples were decalcified with ethylenediaminetetraacetic acid (EDTA), embedded in paraffin, and then sectioned at 4 μm. Conventional, hematoxylin and eosin (H&E) histochemical staining was performed for the assessment of bone sample histology as previously described [33]. For the quantification of the size of adipocytes, five representative sections of each mouse tibia were used. The size of each adipocyte was measured with the use of microscope lenses microscale (Zeiss, Oberkochen, Germany, Axioscope A1).

### 4.3. Whole Bone Marrow Cell Isolation

Under aseptic conditions, tibiae were isolated from C57BL/6 control and *ApoA1* knockout (KO) mice. Whole bone marrow cells were flushed using a 26-gauge syringe filled with cell isolation media (RPMI-1640 with 10% FBS, 1% Pen/Strep (Gibco, Paisley, UK). After red blood cell lysis in ammonium chloride (BD Pharm LyseTM Lysing Buffer, cat no. 555899, BD Biosciences Pharmingen, Billerica, MA, USA) and centrifugation at 20 °C for 5 min, whole bone marrow cells were collected for additional molecular analyses. 

### 4.4. RNA Extraction, cDNA Synthesis, and Real-Time-PCR

Total RNA was extracted from bone marrow cells of WT or *ApoA1*^-/-^ mice tibiae using silica membrane spin columns, NucleoSpin RNA (MACHEREY-NAGEL, Duren, Germany). Extracted RNA was treated with RNase-free DNase to remove contaminating genomic DNA. Total RNA concentration was calculated with a nanodrop spectrophotometer (TECAN, Männedorf, Switzerland). PrimeScript reverse transcriptase (TaKaRa Biotechnology, Shiga, Japan) was used for first-strand cDNA synthesis from total RNA. For real time PCR the MX3000P apparatus (Stratagene, San Diego, CA, USA) was occupied. Primer sets of target and housekeeping genes (Table 1) were from VBC Biotech (Vienna, Austria). Polymerase chain reaction amplification was performed in a final volume of 20 μL including 10 µL of premixed SYBR green, NTPs, buffer, and polymerase (KAPA Biosystems, Boston, MA, USA), plus 20 pmol (1 µL) of each primer and 5 ng (1 μL) of first strand cDNA. The PCR protocol previously published was employed [9]. The primer sets used are presented in Table 1.

### 4.5. Flow Cytometry

For flow cytometry, whole bone marrow cells from four mice of each group were labeled with a cocktail of biotin-conjugated anti-mouse antibodies (CD3, Ly-6G/Ly-6C, CD11b, B220, Ter119, Lineage Panel, Biolegend, San Diego, CA, USA) followed by streptavidin- PerCPCy5.5 (BD Biosciences, Billerica, MA, USA), with anti-mouse PE-CD117 (c-kit) and FITC-Ly-6A/E (Sca-1) (Biolegend, San Diego, CA, USA). At least 20,000 cells were analyzed on a FACSCalibur (BD Biosciences, Billerica, MA, USA) using FlowJo software (Tree Star Inc., Ashland, OR, USA). HSC were defined as Lin-c-kit + Sca1+, common myeloid progenitors as Lin-c-kit + Sca1−, and common lymphoid progenitors as Lin-c-kit low Sca1 low [34]. 

### 4.6. Statistical Analysis

Comparisons were performed using Student’s *t*-test. Data are reported as mean ± standard deviation (mean ± SD). The cut-off point for statistical significance was 0.05 (*p* ≤ 0.05). Analysis was performed using GraphPad Prism 5 (San Diego, CA, USA). 

## Figures and Tables

**Figure 1 ijms-23-04834-f001:**
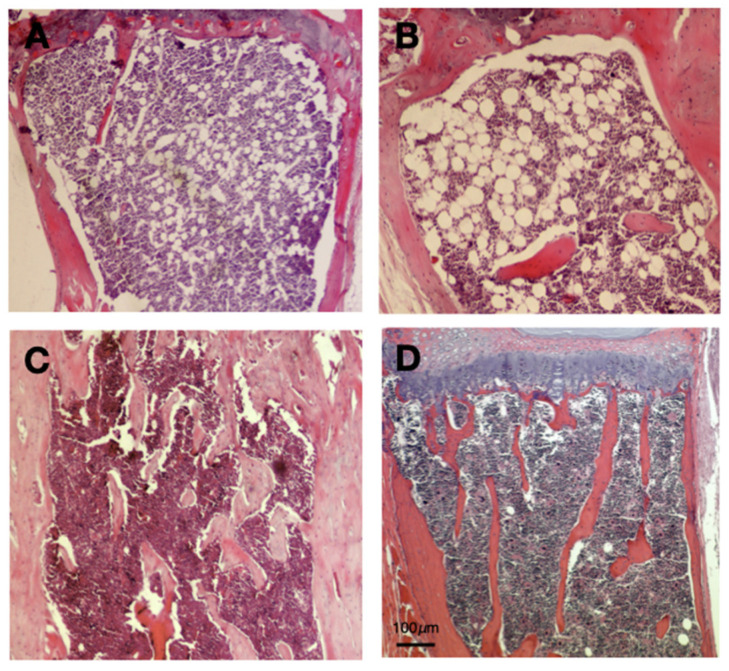
Hematoxylin and eosin sections of *ApoA1* deficient and wild type (WT) mouse tibiae; (**A**,**B**) The bone marrow of *ApoA1* (knockout) KO mice display significant adiposity. The adipocytes are spherical and unilocular, with a single, large lipid vacuole that pushes the nucleus towards the cell periphery; these features are typical of WAT. In addition, these adipocytes are located at the proximal end of tibiae, within red bone marrow; (**C**,**D**) On the contrary, bone marrow of WT mice has very low adiposity. The few adipocytes are similar to those of WAT. Unlike bone marrow fat cells of the *ApoA1* KO mice, their occurrence is random, without any overt tendency for proximal end distribution. Not the reduced number of bone spicules of KO mice tibiae, in line with previous studies [9]. Scale Bar (100 μm) applies to all figures.

**Figure 2 ijms-23-04834-f002:**
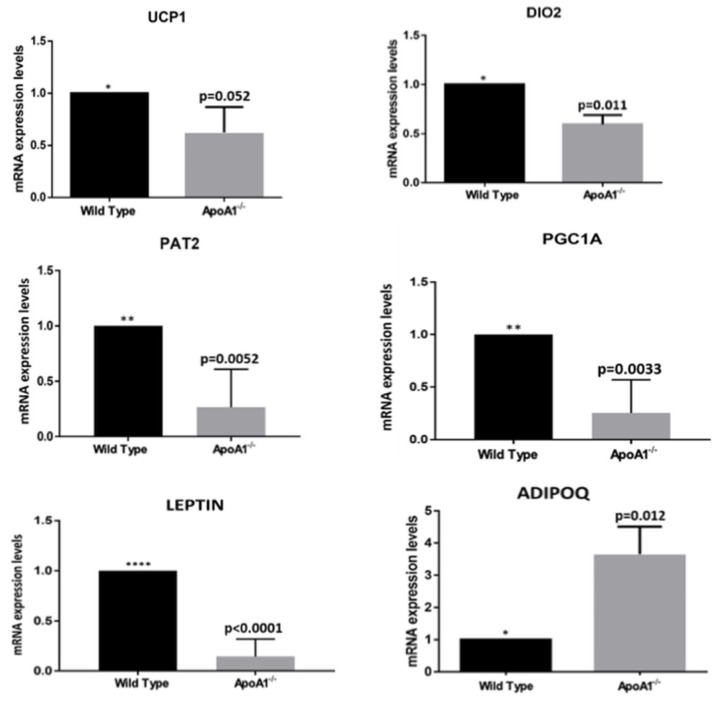
The secondary effects of *ApoA1* deficiency on adipocyte molecular phenotype. All graphs indicate mean ± SD. In all studies, n = 5 for both wild-type and *ApoA1* mRNA isolates. Expression of the brown/beige adipose tissue markers Ucp1, Dio2, Pat2, and Pgc1a were strongly reduced in the *ApoA1* KO mice in comparison to the wild-type counterparts, suggesting a role of APOA1 and probably HDL in the process of bone marrow adipocyte browning. The mRNA of the two major adipokines, leptin and adiponectin, are also shown. Our findings that the mRNA expression of leptin markers is significantly reduced, whereas the mRNA expression of adiponectin (Adipoq) is greatly elevated in the *ApoA1* deficient mice, compared to the controls, is consistent with the hypothesis the APOA1 plays a role in the ‘white-to-brown’ switch in bone marrow adipose tissue. (*: *p* ≤ 0.05, **: *p* ≤ 0.01, ****: *p* ≤ 0.0001).

**Figure 3 ijms-23-04834-f003:**
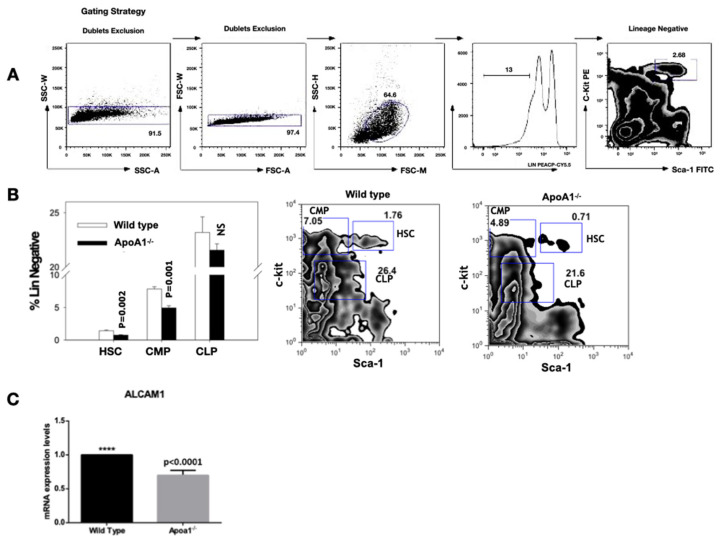
Flow cytometric analysis of whole bone marrow cells. Four animals of each group were tested; (**A**) Gating strategy for hematopoietic stem cells (HSC). The live Lin-population indicated by the bar is further analyzed for c-kit and Sca-1 expression; (**B**) The HSC (Lin-c-kit + Sca1+) and the common myeloid progenitors (CMP; Lin-c-kit + Sca1−) compartment, were significantly reduced in the *ApoA1* KO compared to their wild-type (WT) littermates. On the contrary, the common lymphoid progenitors compartment (CLP; Lin-c-kit low Sca1 low) was unaffected. Data from one representative experiment is shown; (**C**) Reduction in Alcam1 (activated leukocyte cell adhesion molecule 1) expression in the *ApoA1* KO compared to the wild type mice (****: *p* ≤ 0.0001).

**Figure 4 ijms-23-04834-f004:**
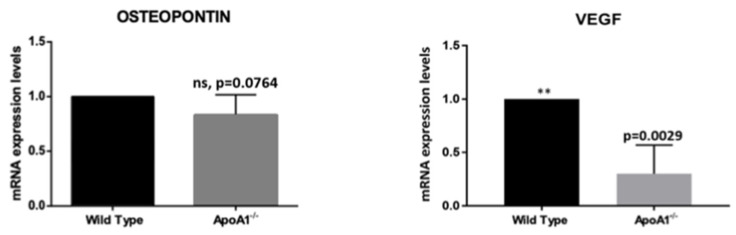
Graphical summaries of mRNA expression of osteopontin (Opn) and VEGFA (Vegfa) in mice. The mRNA of osteopontin (Opn) revealed a strong trend for reduction, without reaching the level of statistical significance. VEGFa expression was greatly reduced in the *ApoA1* deficient mice, consistent with a role of this growth factor in bone marrow adipose tissue ‘beiging’. (**: *p* ≤ 0.01).

**Table 1 ijms-23-04834-t001:** List of primers used for RT-PCR experiments.

PRIMERS	SEQUENCE (5′ -> 3′)
mCEBP f	CGC-AAG-AGC-CGA-GAT-AAA-GC
mCEBP r	CGG-TCA-TTG-TCA-CTG-GTC-AAC-T
mPPAR f	CGC-TGA-TGC-ACT-GCC-TAT-GA
mPPAR r	AGA-GGT-CCA-CAG-AGC-TGA-TTC-C
mAdipoq f	TCA-GTG-GAT-CTG-ACG-ACA-CC
mAdipoq r	AAC-GTC-ATC-TTC-GGC-ATG-ACT
mUCP1 f	TCT-CAG-CCG-GCT-TAA-TGA-CT
mUCP1 r	TGC-ATT-CTG-ACC-TTC-ACG-AC
mDIO2 f	TTC-CTG-GCG-CTC-TAT-GAC-TC
mDIO2 r	TGG-GAG-CAT-CTT-CAC-CCA-GT
mPAT2 f	GGC-TTC-CCA-ACC-ATT-CTG-TC
mPAT2 r	TAC-CGA-CGA-CAT-ACA-GGA-GC
mPGC1a f	ATG-TGT-CGC-CTT-CTT-GCT-CT
mPGC1a r	CGG-TGT-CTG-TAG-TGG-CTT-GA
mLeptin f	CTG-TCT-CCC-ACC-CAT-TCT-GT
mLeptin r	CCA-AGC-CCC-TTT-GTT-CAT-CC
mVEGF f	CCT-GGT-GGA-CAT-CTT-CCA-GGA-GTA-CC
mVEGF r	GAA-GCT-CAT-CTC-TCC-TAT-GTG-CTG-GC
mRBS18 f	GTA-ACC-CGT-TGA-ACC-CCA-TT
mRBS18 r	CCA-TCC-AAT-CGG-TAG-TAG-CG
mOSTEOPONTIN f	GAT-GAT-GAT-GAC-GAT-GGA-GAC-C
mOSTEOPONTIN r	CGA-CTG-TAG-GGA-CGA-TTG-GAG
mPI3K f	AGC-GGA-GAA-CCT-ATT-GCG-AG
mPI3K r	CTT-CGC-CGT-CTA-CCA-CTA-CG
mAKT f	TCA-TTG-AGC-GCA-CCT-TCC-AT
mAKT r	TTC-ATG-GTC-ACA-CGG-TGC-TT
mALCAM f	CCA-TGG-AAC-CGA-TCA-GTG-TGA
mALCAM r	TGC-CGA-CTA-TGC-CAG-TCA-AG

## Data Availability

The datasets used and/or analyzed during the current study are available from the corresponding author on reasonable request. All data generated or analyzed during this study are included in this published article.

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
