# Peer review of "ApoA1 Deficiency Reshapes the Phenotypic and Molecular Characteristics of Bone Marrow Adipocytes in Mice"

_ijms, 2022, doi:10.3390/ijms23094834_

Round 1
Reviewer 1 Report
Using ApoA1-/- mice, this study aims to evaluate the effect of ApoA1 deficiency on the bone marrow adipocytes. I have some major concerns on the study design:
-This is not a conditional KO model, what is the impact of ApoA1 KO on other BM cells apart from adipocytes. Considering ApoA1-/- mice have ore BM adipocytes, one would think that there are relatively fewer hematopoietic cells and/or immune cells in the BM.
-Figure 1: adipocytes are the most abundant cells in the BM. It is unusual to see so few adipocytes in the WT
-Did you quantify the size of adipocytes. Best to show the data.
-Figure: apart from H&E staining, authors should consider staining the adipocytes with adiponectin or perilipin
-Figure 2: It is inappropriate to use total BM cells to study the mRNA profiles of BM adipocytes. The number of BM adipocytes are different between KO and WT mice. There are possibilities that ApoA1-/- changes the composition of BM cells. So, I am unsure how to draw a conclusion from these results.
-Figure 3: It has already been reported that BM adipocytes are negative regulators of HSCs (Naveiras et al Nature 2009).
-There is a lot of discussion about ‘constitutive’ and ‘regulated’ adipocytes in the introduction and discussion sections. It would be useful if the authors investigate the profiles of these BM adipocytes in the absence of ApoA1.
Minor:
-correct typo and grammatical errors
-the intro and discussion need to be more succinct
Author Response
pen Review
Dear Reviewer,
Thank you very much for giving us the opportunity to submit a revision of our review article entitled “ApoA1 deficiency reshapes the phenotypic and molecular characteristics of bone marrow adipocytes in mice” to IJMS.
We would like to thank you for your constructive criticism and the reasonable comments. In the revised version we have made the corrections required. Our changes are highlighted in the revised text with “track changes” tool.
We strongly believe that the revisions that we have made according to your fruitful suggestions have significantly strengthened our paper.
Our point-by-point response to your comments in red font.
Sincerely yours,
Prof. Dionysios Papachristou
Corresponding author
(x) I would not like to sign my review report
( ) I would like to sign my review report
English language and style
( ) Extensive editing of English language and style required
(x) Moderate English changes required
( ) English language and style are fine/minor spell check required
( ) I don't feel qualified to judge about the English language and style
|
Yes |
Can be improved |
Must be improved |
Not applicable |
Does the introduction provide sufficient background and include all relevant references? |
( ) |
(x) |
( ) |
( ) |
Is the research design appropriate? |
( ) |
( ) |
(x) |
( ) |
Are the methods adequately described? |
( ) |
(x) |
( ) |
( ) |
Are the results clearly presented? |
( ) |
(x) |
( ) |
( ) |
Are the conclusions supported by the results? |
( ) |
( ) |
(x) |
( ) |
Comments and Suggestions for Authors
Using ApoA1-/- mice, this study aims to evaluate the effect of ApoA1 deficiency on the bone marrow adipocytes. I have some major concerns on the study design:
-This is not a conditional KO model, what is the impact of ApoA1 KO on other BM cells apart from adipocytes.
Response: We thank the reviewer for this comment. We have previously shown that ApoA1 deficiency results in a significant reduction of BM osteoblasts leading to reduced bone mass. We have also shown that osteoclast number and function remain unaffected (Blair et al, Lab Invest 2016)
Considering ApoA1-/- mice have ore BM adipocytes, one would think that there are relatively fewer hematopoietic cells and/or immune cells in the BM.
Response: This is a reasonable comment that is line with our finds that ApoA1 deficiency results in a significant reduction of HSC and common myeloid progenitors. Common lymphoid progenitors are not affected. However, as mentioned in the paper, these findings are descriptive and need further investigation. This is stated clearly in the revised version of our paper.
-Figure 1: adipocytes are the most abundant cells in the BM. It is unusual to see so few adipocytes in the WT
Response: The reviewer is right. However, these mice are young (12w) and adipocytes are a very small fraction of the bone marrow cells. This has been reported to previous studies of our research group and proved by both in histological and in vitro methods (Blair et al, Lab Invest 2016). However, in oder to better address the reviewer’s point, we added more microphotographs of BM from ApoA1 KO and WT mice (Figure1) of the revised paper.
-Did you quantify the size of adipocytes. Best to show the data.
Response: We thank the reviewer for this well-received comment. We added this statistical data in the MM and results section of revised MS.
-Figure: apart from H&E staining, authors should consider staining the adipocytes with adiponectin or perilipin
Response: This is also a right comment. In previous related publication we have made histological and histomorphometric analyses on BM adipocytes of ApoA1 knock out and C57BL/6 WT mice (Blair et al, Lab Invest 2016) (see below).
Figure2: It is inappropriate to use total BM cells to study the mRNA profiles of BM adipocytes. The number of BM adipocytes are different between KO and WT mice. There are possibilities that ApoA1-/- changes the composition of BM cells. So, I am unsure how to draw a conclusion from these results.
Response: The reviewer is right as regards the limitations of the use of WBMC for studying the profile of adipocytes. Nevertheless, this methodology has been successfully used in previous publications (Blair et al, Lab Invest 2017, Tourkova et al, Lab Invest 2019) and the results exported from WMBC were similar to those of the in vitro experiments. In addition, the primers used are very specific for white or brown adipocyte-related genes.
However, if necessary we could differentiate adipocytes in vitro with and without ApoA1, and then test these for “beige” changes. Nonetheless, this would take about 6 months and the authors do not feel this is justified.
-Figure 3: It has already been reported that BM adipocytes are negative regulators of HSCs (Naveiras et al Nature 2009).
Response: We thank the reviewer for this comment. We added this reference and made changes in the “Discussion” section, accordingly.
-There is a lot of discussion about ‘constitutive’ and ‘regulated’ adipocytes in the introduction and discussion sections. It would be useful if the authors investigate the profiles of these BM adipocytes in the absence of ApoA1.
Response: This is a very interesting suggestion, indeed. However, the isolation of BM abdicates is practically impossible under the condition this study has been designed. We could start a project like this is a set of new ApoA1 KO mice but this this would take about 1 year and the authors do not feel this is justified.
Minor:
-correct typo and grammatical errors
Response: We have done the corrections needed
-the intro and discussion need to be more succinct
Response: We have made changes is the “Introduction” and “Discussion” section that have improved the quality of the paper, according to the reviewer’s suggestions.
Submission Date
18 March 2022
Date of this review
22 Mar 2022 15:43:16
Reviewer 2 Report
The study presented by Kastrenopoulou and colleagues brings some novel and important results and assumtions. The topic of the study is quite relevant and actual. However, in my opinion due to missing clear evidences regarding marrow adipocytes at cellular levels, I strongly suggest to the authors to re-consider their results and in that context to mitigate their conclusion on "reshapes the phenotypic and molecular characteristics of bone marrow adipocytes".
Major comments:
- Can authors more directly define marrow adipocyte profile instead of mRNA analyses of bulk bone marrow cell populations? If not, I would rather re-write these results parts since it mostly describes the impact of ApoA1 deficiency on the entire marrow transcriptome, rather than marrow adipocytes. Particularly, it is very imprtant to be specific if want to define white or beige adipocyte phenotype.
- Evidences on hematopoietic compartmants would be necessary to improve. Authors analyzed some surface markers without showing functionality of HSPC. Markers analyzed and presented here are simply not strong enough to define CMP or CLP populations. So, please rewrite results part regarding this and be as specific as possible.
Author Response
pen Review
Dear Reviewer,
Thank you very much for giving us the opportunity to submit a revision of our review article entitled “ApoA1 deficiency reshapes the phenotypic and molecular characteristics of bone marrow adipocytes in mice” to IJMS.
We would like to thank you for your constructive criticism and the reasonable comments. In the revised version we have made the corrections required. Our changes are highlighted in the revised text with “track changes” tool.
We strongly believe that the revisions that we have made according to your fruitful suggestions have significantly strengthened our paper.
Our point-by-point response to your comments in red font.
Sincerely yours,
Prof. Dionysios Papachristou
Corresponding author
Open Review
(x) I would not like to sign my review report
( ) I would like to sign my review report
English language and style
( ) Extensive editing of English language and style required
( ) Moderate English changes required
(x) English language and style are fine/minor spell check required
( ) I don't feel qualified to judge about the English language and style
|
Yes |
Can be improved |
Must be improved |
Not applicable |
Does the introduction provide sufficient background and include all relevant references? |
( ) |
(x) |
( ) |
( ) |
Is the research design appropriate? |
( ) |
( ) |
(x) |
( ) |
Are the methods adequately described? |
( ) |
( ) |
(x) |
( ) |
Are the results clearly presented? |
( ) |
( ) |
(x) |
( ) |
Are the conclusions supported by the results? |
( ) |
( ) |
(x) |
( ) |
Comments and Suggestions for Authors
The study presented by Kastrenopoulou and colleagues brings some novel and important results and assumtions. The topic of the study is quite relevant and actual. However, in my opinion due to missing clear evidences regarding marrow adipocytes at cellular levels, I strongly suggest to the authors to re-consider their results and in that context to mitigate their conclusion on "reshapes the phenotypic and molecular characteristics of bone marrow adipocytes".
Response: We have made changes in the Discussion section of our revised MS according to the reviewer’s well-received suggestion.
Major comments:
- Can authors more directly define marrow adipocyte profile instead of mRNA analyses of bulk bone marrow cell populations? If not, I would rather re-write these results parts since it mostly describes the impact of ApoA1 deficiency on the entire marrow transcriptome, rather than marrow adipocytes. Particularly, it is very imprtant to be specific if want to define white or beige adipocyte phenotype.
Response: We thank the reviewer for this fair comment as regards the use of WBMC for studying the profile of adipocytes. Nevertheless, this methodology has been successfully used in previous publications (Blair et al, Lab Invest 2017, Tourkova et al, Lab Invest 2019) and the results exported from WMBC were similar to those of the in vitro experiments. In addition, the primers used are specific for white or brown adipocytes.
However, if necessary we could differentiate adipocytes in vitro with and without ApoA1, and then test these for “beige” changes. Nonetheless, this would take about 6 months and the authors do not feel this is justified.
- Evidences on hematopoietic compartmants would be necessary to improve. Authors analyzed some surface markers without showing functionality of HSPC. Markers analyzed and presented here are simply not strong enough to define CMP or CLP populations. So, please rewrite results part regarding this and be as specific as possible.
Response: This is a very significant comment. However, this study is descriptive and did not aim at studying the functionality of HSC or the molecular mechanisms involved in the regulation of apoA1 deficiency on HSC and subpopulations. However, in the revised version of our paper we commented this following the reviewers suggestions, which we think, have significantly improved the quality of our paper.
Submission Date
18 March 2022
Date of this review
02 Apr 2022 20:44:12as
Round 2
Reviewer 1 Report
Lin- c-kit + Sca1+ refers to the hematopoietic stem cell compartment not the hematopoietic stem cell.
Author Response
Dear Reviewer,
Thank you very much for your fair comment as regards the HSC compartment. It is true that is not possible to identify actual HSCs, as is widely appreciated. Following your suggestion, in the second revision of our paper (R2), we have used the term “compartment” at the “abstract”, “results” and “materials and methods” sections. Our changes are highlighted in the revised text with “track changes” tool.
We strongly believe that the revisions that we have made according to your fruitful suggestions have significantly strengthened our paper.
Sincerely yours,
Prof. Dionysios Papachristou
Corresponding author
Reviewer 2 Report
The Authors mostly improved the paper. It can be considered for publication.
Author Response
Dear Reviewer,
Thank you very much for accepting our paper for publication at IJMS.
We strongly believe that the revisions that we have made according to your fruitful suggestions have significantly elevated the quality of our paper.
Sincerely yours,
Prof. Dionysios Papachristou
Corresponding author